# Multimodal Masked Autoencoders Learn Transferable Representations

## Abstract

Building scalable models to learn from diverse, multimodal data remains an open challenge. For vision-language data, the dominant approaches are based on contrastive learning objectives that train a separate encoder for each modality. While effective, contrastive learning approaches introduce sampling bias depending on the data augmentations used, which can degrade performance on downstream tasks. Moreover, these methods are limited to paired image-text data, and cannot leverage widely-available unpaired data. In this paper, we investigate whether a large multimodal model trained purely via masked token prediction, without using modality-specific encoders or contrastive learning, can learn transferable representations for downstream tasks. We propose a simple and scalable network architecture, the Multimodal Masked Autoencoder (M3AE), which learns a unified encoder for both vision and language data via masked token prediction. We provide an empirical study of M3AE trained on a large-scale image-text dataset, and find that M3AE is able to learn generalizable representations that transfer well to downstream tasks. Surprisingly, we find that M3AE benefits from a higher text mask ratio (50-90%), in contrast to BERT whose standard masking ratio is 15%, due to the joint training of two data modalities. We also provide qualitative analysis showing that the learned representation incorporates meaningful information from both image and language. Lastly, we demonstrate the scalability of M3AE with larger model size and training time, and its flexibility to train on both paired image-text data as well as unpaired data.

## 1 Introduction

With the rapid advances in neural architectures (Vaswani et al., 2017) and hardware performance, self-supervised pre-training has made tremendous progress in natural language processing (NLP) and vision (He et al., 2021; Devlin et al., 2018; Bao et al., 2021; Brown et al., 2020). The underlying idea, often referred as masked token prediction, is conceptually simple: the model learns to predict a removed portion of the data. Masked token prediction has enabled highly successful methods for pre-training in NLP and vision, including Transformer (Vaswani et al., 2017), GPT (Brown et al., 2020), BERT (Devlin et al., 2018), and MAE (He et al., 2021). These pre-trained representations have been shown to generalize well to various downstream tasks. The cornerstone of these successes is that these methods excellently leverage large and diverse datasets. Indeed, with the scaling up of data diversity and model capacity, there is still no sign of plateau on generalization to various downstream tasks (Devlin et al., 2018; He et al., 2021).

Driven by the successes in NLP and vision, there has been significant interest in improving visual representation learning by training on large and diverse multimodal datasets that contains both images and text. These datasets, such as CC12M (Changpinyo et al., 2021) and YFCC100M (Thomee et al., 2015), are often much more scalable than explicitly labeled datasets such as ImageNet (Deng et al., 2009), and the diverse language data can provide rich supervision to train more generalizable representations.

The dominant paradigm for multimodal pre-training is cross-modal contrastive learning, such as CLIP (Radford et al., 2021) and ALIGN (Jia et al., 2021). These methods show that cross-modal contrastive learning models, trained on giant corpora of paired image-and-text, can generalize well to various downstream tasks. Despite these progresses, a major limitation for contrastive learning is that

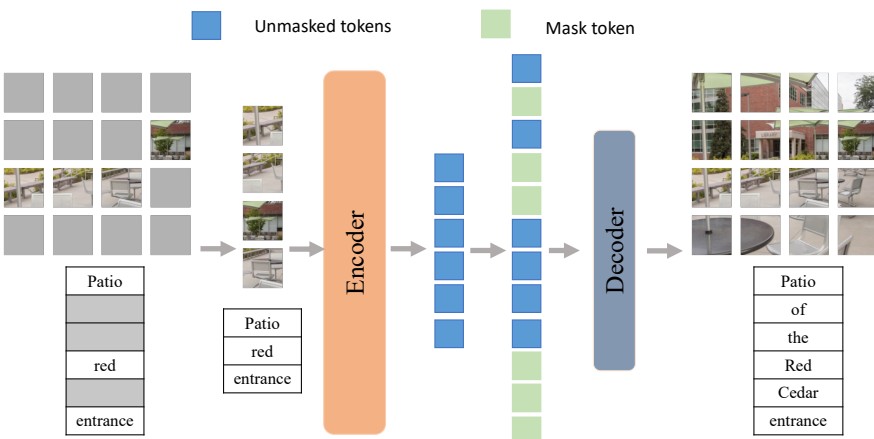

**Figure 1:** Multimodal masked autoencoder (M3AE) consists of an encoder that maps language tokens and image patches to a shared representation space, and a decoder that reconstructs the original image and language from the representation.

it requires paired image-and-text data and therefore cannot leverage widely available unpaired data. In addition, contrastive learning based methods use separate encoders for image and text, making it difficult for models to access information from different modalities at the same time. The separation of image and text encoders hinder the joint understanding of image and text.

To address the above limitations for visual representation learning, we propose a simple and scalable architecture called the multimodal masked autoencoders (M3AE) for learning a single unified model on large image and language data, without using modality-specific encoders or contrastive learning. Based on MAE (He et al., 2021), M3AE is trained purely via masked token prediction. Our key idea is to treat an image-and-text pair as a long sequence of tokens consisting of embeddings of image patches and text. M3AE is trained simply by masking random patches of the input image and language tokens, and learning to reconstruct the masked pixels and text.

In this paper, we provide an empirical study of M3AE trained on the multimodal CC12M (Changpinyo et al., 2021) dataset, and find that M3AE is able to learn generalizable representations that transfer well to downstream tasks such as image classification and out-of-distribution detection. We find that multimodal pre-training of M3AE on CC12M achieves significantly higher performance on the ImageNet-1k linear classification benchmark (Russakovsky et al., 2014) compared to pre-training on images only (MAE). Our strong results for M3AE demonstrate the generalization benefits of multimodal training for learning transferable representations across datasets.

Surprisingly, we find that M3AE performs best when we apply a high mask ratio (75%) on language, while in contrast, language models like BERT (Devlin et al., 2018) conventionally use a low mask ratio (15%) because language data are highly semantic and information-dense. We hypothesize that M3AE benefits from a higher mask ratio on text because it enforces a better joint understanding of vision and language during masked token prediction. We also provide qualitative analysis showing that the learned representation incorporates meaningful information from both image and language. Furthermore, we demonstrate the scalability of M3AE with larger model size and training time, as well as its flexibility to train on both paired image-text data as well as unpaired data.

## 2 RELATED WORK

**Self-supervised representation learning via reconstruction**    After the introduction of Transformers (Vaswani et al., 2017), self-supervised language modeling has made substantial progress in recent years. After pre-training on a large amount of unlabeled data with reconstruction loss, Large-scale transformer language models like BERT (Devlin et al., 2018) and GPT (Brown et al., 2020) are highly successful in learning representations that generalize well to various downstream tasks. Taking inspiration from the success in NLP, research have proposed a wide variety of self-supervision method (Chen et al., 2020a; Dosovitskiy et al., 2020; Bao et al., 2021; He et al., 2021). iGPT (Chen et al., 2020a) that operates on sequences of pixels and reconstruct the unknown pixels. ViT (Doso-

vitskiy et al., 2020) studies masked patch prediction for self-supervised learning. BEiT (Bao et al., 2021) proposes to predict discrete tokens (Van Den Oord et al., 2017; Ramesh et al., 2021). MAE (He et al., 2021) proposes to randomly mask patches of the input image and reconstruct the missing pixels. Heavily inspired by MAE and BERT, our M3AE brings together image and language data and learns a shared representation for both modalities by applying a unified masked patch and token prediction objective.

**Self-supervised representation learning via contrastive objectives** Besides reconstruction, another major paradigm for self-supervised learning is contrastive learning, which models similarity and dissimilarity between two or more views of images or texts (Gao et al., 2021; Chen et al., 2020c; He et al., 2020; van den Oord et al., 2018; Grill et al., 2020; Wu et al., 2018). SimCSE (Gao et al., 2021) proposes constructing positive sentence pair through applying Dropout. SimCLR (Chen et al., 2020b) studies applying random image augmentation for contrastive learning. Contrastive learning often rely heavily on data augmentation and can therefore introduce bias during training. Our M3AE does not rely on contrastive objectives so it can be applied without data augmentation.

**Joint learning for language and image** Learning representations for a single modality has high importance as it extracts semantic formation from the raw data of modality. Learning a joint representation for several modalities is challenging since it requires alignment between semantic information from different modalities, of which the information contained may vary drastically. Specifically, learning joint representation for vision and language has been a long standing problem in artificial intelligence. Recently, CLIP (Radford et al., 2021) successfully tackled this challenge by leveraging contrastive learning over a large dataset of aligned text-image pairs. Several works followed this idea, further improving the joint representation. BLIP (Li et al., 2022) used noisy web data by bootstrapping the captions with synthetic ones. SLIP (Mu et al., 2021) learned a joint representation by combining CLIP (Radford et al., 2021) and SimCLR (Chen et al., 2020b) techniques and leveraging both a paired dataset, and a much larger image-only dataset. DeCLIP (Li et al., 2021) utilized more image-text pairs collected from CLIP (Radford et al., 2021) by adding multiple self-supervised techniques. Inspired by BERT, other methods study cross-modal matching loss (Chen et al., 2019; Lu et al., 2019; Singh et al., 2021; Tan & Bansal, 2019; Yu et al., 2022). FLAVA (Singh et al., 2021) employs both contrastive and multimodal training objectives on paired and image-only datasets. Perceiver (Jaegle et al., 2021) proposes cross-attention to combining language and image modalities. CoCa (Yu et al., 2022) combines cross-modal contrastive learning and autoregressive caption prediction. Our M3AE models provides a simple but effective alternative for learning joint representations by processing language tokens and image patches through a shared encoder-decoder architecture. We train our model with masked token reconstruction loss, eliminating the need to handle each modality separately.

## 3 MULTIMODAL MASKED AUTOENCODER (M3AE)

In this section we introduce our method, multimodal masked autoencoder (M3AE), which consists of an encoder that maps image and language to representation space, and a decoder that reconstructs the original image and language from the representation. We summarize the main architecture and training process of M3AE in Figure 1 and Figure 2.

**Image-language masking.** The first step of M3AE is to combine the language and image input into a single sequence. Following standard natural language processing practice (Devlin et al., 2018), we tokenize the input text into a sequence of discrete tokens. For image input, we divide it into regular non-overlapping patches of pixels, following the practice of ViT (Dosovitskiy et al., 2020). Text tokens and image patches are then concatenated into a single sequence.

For patches and tokens, we sample s random subset without replacement from a uniform distribution, and mask (*i.e.*, remove) the remaining ones. A *high* masking ratio is applied to both text tokens and image patches, in order to eliminate information redundancy and make a sufficiently difficult task that cannot be easily solved by extrapolation from visible neighboring patches and tokens.

**M3AE encoder.** The M3AE architecture consists of two networks: an encoder and a decoder. The encoder is a large transformer, following the architecuture of ViT (Dosovitskiy et al., 2020) and

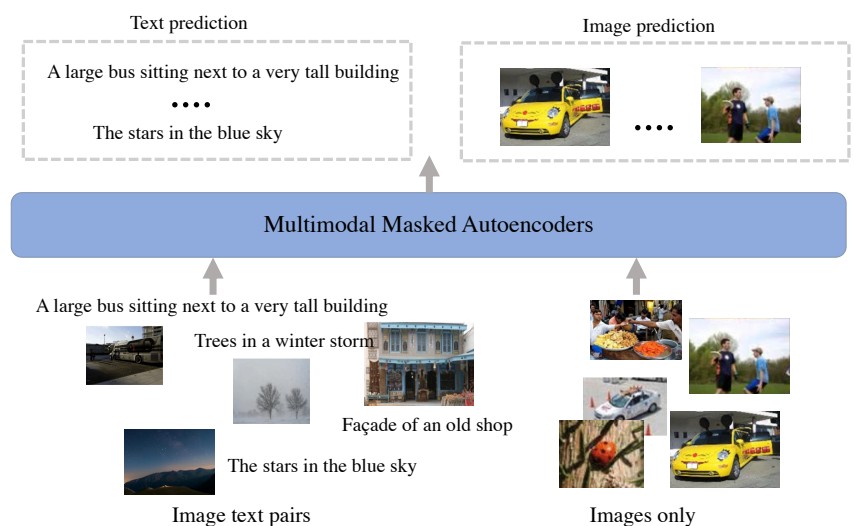

**Figure 2:** M3AE can learn representations from a flexible mixture of image-text pairs and unpaired images using a unified model without relying on data augmentations.

BERT (Devlin et al., 2018). The encoder takes only *unmasked (visible)* language tokens and image patches as input. For language tokens, we first convert it into learnable embedding vectors and then apply 1D positional encodings, following the standard practice (Devlin et al., 2018). For image patches, we use a learnable linear projection to convert them to image embeddings that have the same dimension as the language embeddings, and then apply 2D positional encodings, following the practice of MAE He et al. (2021). In order to distinguish the two different modalities, we add two learnable vectors that represent language and images respectively to the corresponding modalities' embeddings. We call these "modality type encodings". Additionally, a learnable CLS embedding (Devlin et al., 2018) is prepended to the beginning of the sequence. The combined language and image embeddings are then processed by a series of transformer blocks to obtain the final representation. Although the input consists of long sequences of image patches and text tokens, we can still train very large transformer encoders efficiently because the same only operates on a small subset (*e.g.*, 25%) of the full set.

**M3AE decoder.** Following MAE (He et al., 2021), we use a lightweight transformer-based decoder on the full set of tokens consisting of (i) encoded visible image patches, (ii) encoded visible text tokens, and (iii) mask tokens. Each mask token is a shared, learned vector that indicates the presence of a missing patch or token to be predicted. We add positional embeddings to all tokens in this full set in order to encode location information in mask tokens. We also add a different set of modality type embeddings to visible tokens, similar to the encoder. After the decoder transformer, we apply two linear projection output heads to compute the reconstruction. The image output head projects the decoder output corresponding to image patches to the same dimension as pixels in the original image patches. The language output head projects the decoder output of language to token logits. These output heads are then used for supervision during the self-supervised training of M3AE.

**Self-supervised training of M3AE.** Our M3AE reconstructs the input by predicting the *pixel* values for masked image patches and the token probabilities for masked language tokens. For image reconstruction, we compute the mean squared error (MSE) between the reconstructed and original images in the pixel space. For language reconstruction, we apply the cross entropy loss between the reconstructed and original text. Our loss is a weighted sum of the image loss and the text loss. Similar to MAE (He et al., 2021) and BERT (Devlin et al., 2018), we compute the loss only on the masked image patches and language tokens. Since M3AE processes image and language data uniformly by combining them into a single sequence, a natural advantage for our model is that it can be trained with the exact same loss on a mixture of paired and unpaired data as shown in Figure 2, significantly extending the applicability of our model beyond what is possible with contrastive learning.

## 4 EXPERIMENTS

In this section, we study the representation quality of M3AE. We aim to answer the following questions in our experiments: **(1)** Can M3AE learn generalizable visual representations that transfer well to downstream tasks? **(2)** Does the learned representation incorporate meaningful information from both images and language? **(3)** Does M3AE scale well with model size and training time?

To answer these questions, we first pre-train the M3AE model on a diverse image-and-language dataset and evalute its performance for downstream classification and out-of-distribution detection. We further evaluate the scalability of the model with respect to training epochs and model size. Finally, we provide a detailed ablation study and qualitative analyses to inspect the quality of the learned representations.

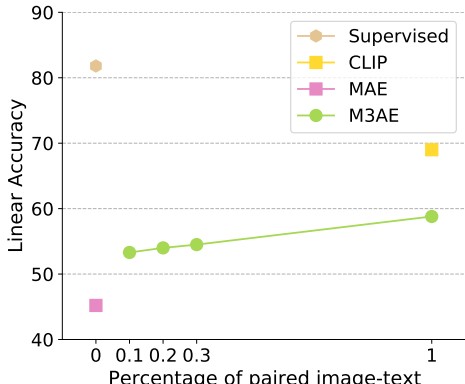

| Model | MAE | M3AE | CLIP | Supv |
|---|---|---|---|---|
| Accuracy | 44.6 | 61.3 | 69.0 | 81.8 |
| M3AE text ratio | 10% | 20% | 30% | 100% |
| Accuracy | 53.3 | 54.0 | 54.5 | 58.8 |

**Figure 3:** Comparison of M3AE, MAE, and CLIP on ImageNet. M3AE significantly outperforms MAE. M3AE can flexibly leverage a combination of paired image-text data and unpaired image only data. All models are ViT-B. MAE and M3AE are pretrained on CC12M for 50 epochs.

## 4.1 DATASETS

**Pre-training datasets.** M3AE is trained on Conceptual 12M (CC12M) (Changpinyo et al., 2021). The original dataset images are provided in the form of internet URLs. Note that due to some expired URLs and non-English captions, we did not obtain the complete data in the dataset. For language data, we use the BERT tokenizer from Huggingface[1] to tokenize the text. We provide more details about data preprocessing in Section A.1.

**Downstream datasets.** We assess model performance in a wider variety of distributions and tasks. We evaluate the image encoder transferability on ImageNet (Russakovsky et al., 2014). We report top-1 validation accuracy of a single 224×224 crop. We evaluate out-of-distribution detection on CIFAR-100 and CIFAR-10 datasets (Krizhevsky et al., 2009).

## 4.2 EXPERIMENTS SETUP

**Network architectures.** Following MAE, we use ViT (Dosovitskiy et al., 2020) as the model architecture and consider three different sizes of ViT for the M3AE image and text encoder. We use the original ViT-B/16 and ViT-L/16 architectures (Dosovitskiy et al., 2020) for our encoder, as well as ViT-S/16 (Touvron et al., 2021) which is comparable to ResNet-50 in FLOPs and parameters. Following MAE (He et al., 2021), our decoder is lightweight and has 8 blocks and a width of 512. Full details about network architectures can be found in Section A.3.

**Pre-training setup.** For a fair comparison with MAE, we train our model from scratch for the same number of epochs as MAE. The learnable temperature parameter $\tau$ is initialized to 0.01. The loss weights of image prediction and text prediction are set to 1 and 0.5. The mask ratio for image and text are both set to 0.75. Refers to Section A.4 for more details.

**Downstream evaluation setup.** We evaluate our model transferability by performing linear classification on frozen features, *i.e.*, the pre-trained image encoder is fixed and serves as a feature extractor. After feature extraction, we train the linear classifier with the AdamW (Loshchilov & Hutter, 2018) optimizer as the same in He et al. (2021). More details can be found in Section A.5

## 4.3 RESULTS

**ImageNet Linear Classification.** We evaluate performance on ImageNet under the linear classification setting. Linear classification, also called linear probing, is a standard evaluation method used to

---

[1]https://huggingface.co/docs/transformers/main_classes/tokenizer

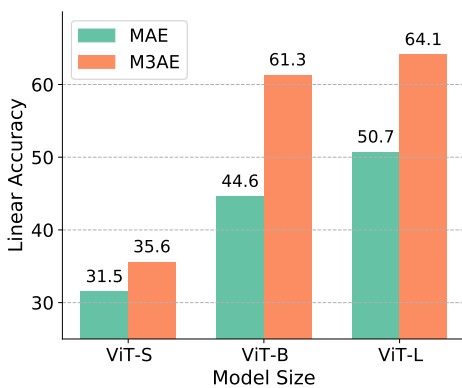
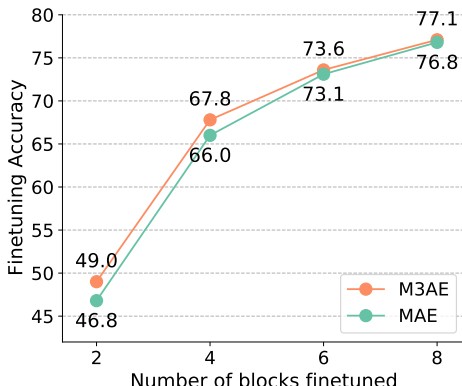

**Figure 4: Left**: Comparing the linear classification accuracy ViT model variants of different capacities (ViT-S/B/L). All models are pre-trained for 50 epochs. M3AE scales well with model size, outperforming MAE in every setting. **Right**: Comparing finetuning different number of blocks for ViT-L. All models are pre-trained for 50 epochs.

evaluate unsupervised or self-supervised representations. A randomly initialized final classification layer is trained while all other model weights are frozen.

Figure 3 shows the results of linear classification. We report the results of ViT-B trained on ImageNet and CLIP (Radford et al., 2021) pre-trained on CC12M from prior work (Touvron et al., 2021; Mu et al., 2021).To study the flexibility of M3AE, we remove the text for a portion of image-text pairs, *i.e.*, 30% of paired image-text examples means 70% of CC12M image-text pairs become images only. A lower percentage of paired image-text data contains less information and therefore makes the task more difficult, since the model has to infer the relation between visual and language concepts based on limited paired data.

The comparison between M3AE and the baselines are shown in Figure 3. M3AE significantly outperforms MAE by nearly 10 percent. CLIP is a strong baseline based on cross-modal contrastive learning. While it achieves higher accuracy than M3AE, it is less flexible than our model since it can only use paired image-text data. In contrast, M3AE can leverage both paired image-text and unpaired image data without modifying the training procedure, as shown in Figure 3, giving our model strong potential to leverage a diverse combination of unpaired single modality and multi-modal data. Notably, with M3AE pre-training, even adding 10% text annotations leads to a significant boost in accuracy over MAE (53.3% vs 45.2%).

We make an important note that the linear classification performance of MAE pre-trained on CC12M is much lower than MAE pre-trained on ImageNet, and we hypothesize that such a difference is caused by the *large domain gap between the two datasets*. To

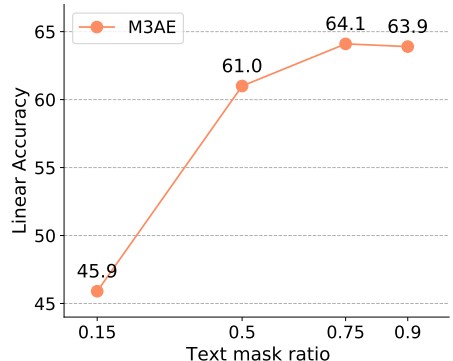

**Figure 5:** Comparing M3AE with different text mask ratio. All models are ViT-L trained for 50 epochs on CC12M. We see that M3AE performs the best with a surprisingly high text mask ratio of 75%.

confirm this hypothesis, we pre-trained a ViT-L MAE on ImageNet for 800 epochs using the same hyperparameters on top of our implementation, and obtained 73.5% accuracy on linear classification, which exactly matches the original reported performance (He et al., 2021). Thus, while our results cannot be directly compared to the original MAE results (He et al., 2021) pre-trained on ImageNet due to distribution mismatch, they demonstrate the strengths of multimodal training of M3AE for learning transferable representations across datasets.

**ImageNet Fine-tuning.** Following the fine-tuning experiment settings of MAE (He et al., 2021) , we perform partial fine-tuning on pretrained models: fine-tune the last several layers while freezing

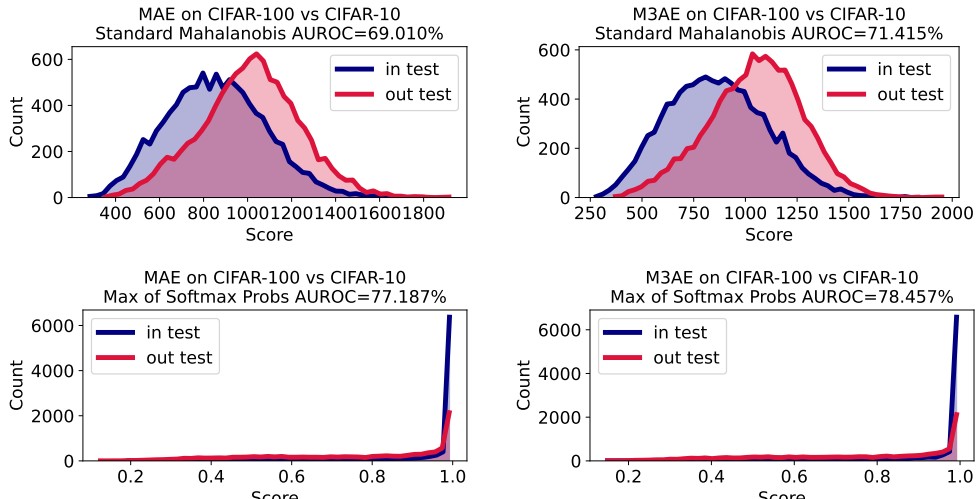

**Figure 6:** Out-of-distribution detection results on CIFAR-100 (in-distribution) and CIFAR-10 (out-of-distribution). **Upper** shows results based on Mahalanobis outlier score, M3AE achieves *71.4%* which is higher than MAE's *69.0%*. **Lower** shows results based on max over softmax score, M3AE achieves *78.5%* which is also higher than MAE's *77.2%*.

the others. Figure 4 (right) shows that M3AEoutperforms MAE under different number of finetuned layers. This results suggest that M3AE representations are more separable than MAE, which are also visualized in Section 4.4. We notice that with more layers being finetuned, the gap between M3AE and MAE becomes smaller. We believe the reason is that there is a large domain gap between internet image-text datasets and ImageNet images, therefore finetuning more layers may essentially destroy pretrained representations. Nonetheless, M3AE learns highly transferable representations that performs well even with the large domain gap between CC12M and ImageNet.

## 4.4 ANALYSIS

**Model scaling.** We also investigate the scaling behavior of M3AE with larger vision Transformer models. We pre-train M3AE and MAE with ViT-Small, ViT-Base and ViT-Large and perform ImageNet linear classification and finetuning using the learned representations. Figure 4 shows the effect of different model sizes. Our results indicate that M3AE scales well larger models, significantly outperforming MAE across different model sizes.

**Out-of-distribution detection.** Some prior work demonstrated self-supervised learning approaches significantly improve OOD detection performance (Hendrycks et al., 2019; 2020; Fort et al., 2021), where their self-supervised pre-training heavily relies on domain-specific data augmentations. We expect MAE to perform well on OOD benchmarks and want to study how M3AE performs compared with MAE.

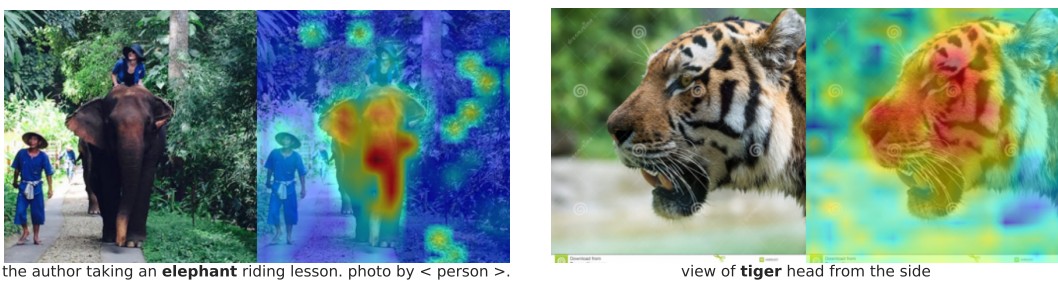

the author taking an **elephant** riding lesson. photo by < person >.          view of **tiger** head from the side

**Figure 7:** Visualization of attention between a given text token and image patches on CC12M dataset. The text token for which we visualize the attention is bolded. We see that the M3AE encoder is able to attend to the correct objects.

We consider the difficult near-OOD as this is a more challenging and realistic problem; many methods can achieve high AUROC on the easier far-OOD benchmarks, but do not perform as well in near-OOD tasks. The results are shown in Figure 6, M3AE outperforms MAE in terms of both Mahalanobis outlier score (Lee et al., 2018) and max over softmax score (Hendrycks et al., 2018).

**Ablation on text mask ratio.** We also investigate the performance of M3AE under various text mask ratios. Figure 5 shows holding the image patch mask ratio fixed (75%) and training for various text mask ratios. Surprisingly, the results indicate that M3AE benefits from a high text mask ratio (50%-90%), contrary to BERT (Devlin et al., 2018) whose typical masking ratio is 15%. We believe that this is the result of joint training of two modalities of data, where the masked language prediction can make use of information from both the visible language tokens and image patches.

**Visualization of cross-modal attention weights.** We are interested in what M3AE captures in multimodal attention weights. To do so, we visualize the M3AEencoder attention between a given text token and all image patches, as well as the attention between a given image patch and all text tokens (Yang & Zhang, 2018) in Figure 7 and Figure 8. M3AE learns to attend relevant concepts in both image and text, showing that our model is able to infer relations between visual and language concepts.

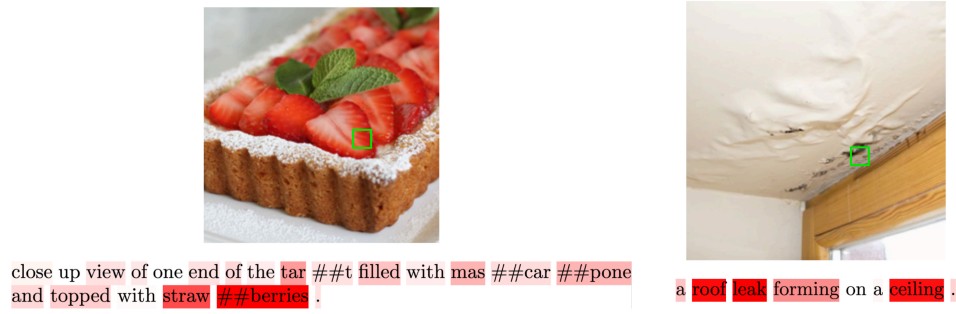

close up view of one end of the tar ##t filled with mas ##car ##pone and topped with straw ##berries .

a roof leak forming on a ceiling .

**Figure 8:** Visualization of attention between a given image patch and all text tokens on CC12M dataset The highlighted rectangle is the image patch for which we visualize the attention. Denser color of the text denotes higher attention. The visualization suggests that M3AE encoder is able to attend to the correct words corresponding to the image patch.

**Reconstruction visualization.** We are interested in the reconstruction quality of pretrained M3AE. We randomly sample examples from CC12M and the validation set of ImageNet and show the results in Figure 9 and Figure 10. In each reconstructed image, we include original unmasked tokens for better visual quality. We observe that our model infers holistic reconstructions across CC12M and ImageNet datasets, indicating it has learned numerous concepts.

**Clustering analysis of representation.** We perform t-SNE (Van der Maaten & Hinton, 2008) visualizations of the learned representation of M3AE and MAE for 10 classes on ImageNet validation

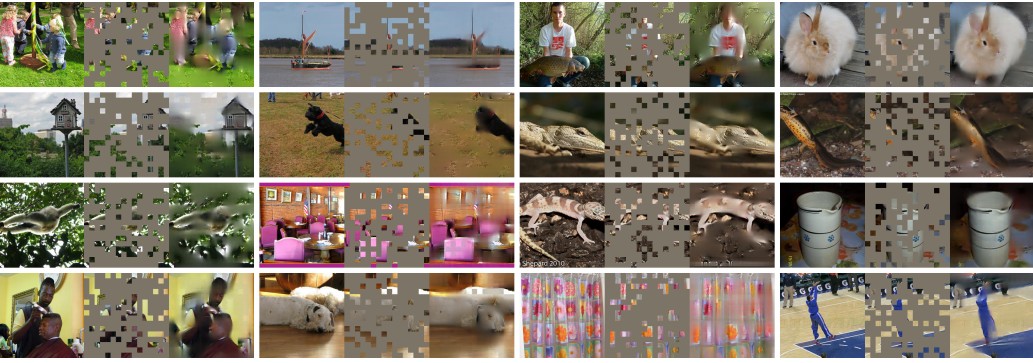

**Figure 9:** Masked image reconstruction on ImageNet validation images. For each triplet, we show the ground-truth (left), the masked image (mid) and our M3AE reconstruction (right).

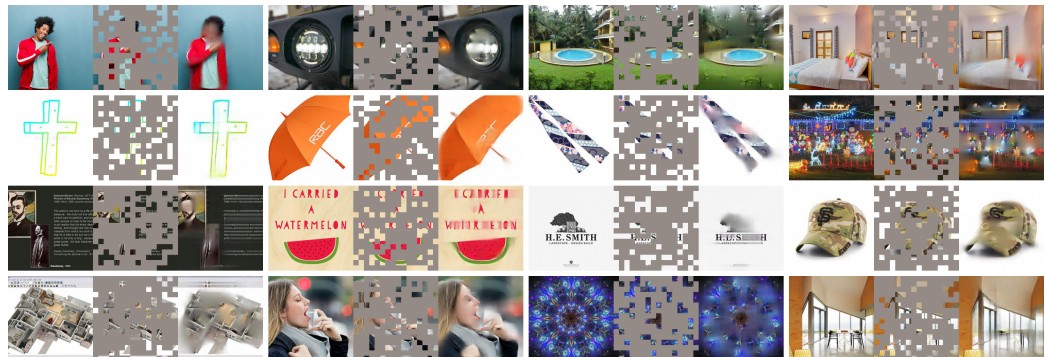

**Figure 10:** Masked image reconstruction on CC12M images. For each triplet, we show the ground-truth (left), the masked image (mid) and our M3AE reconstruction (right).

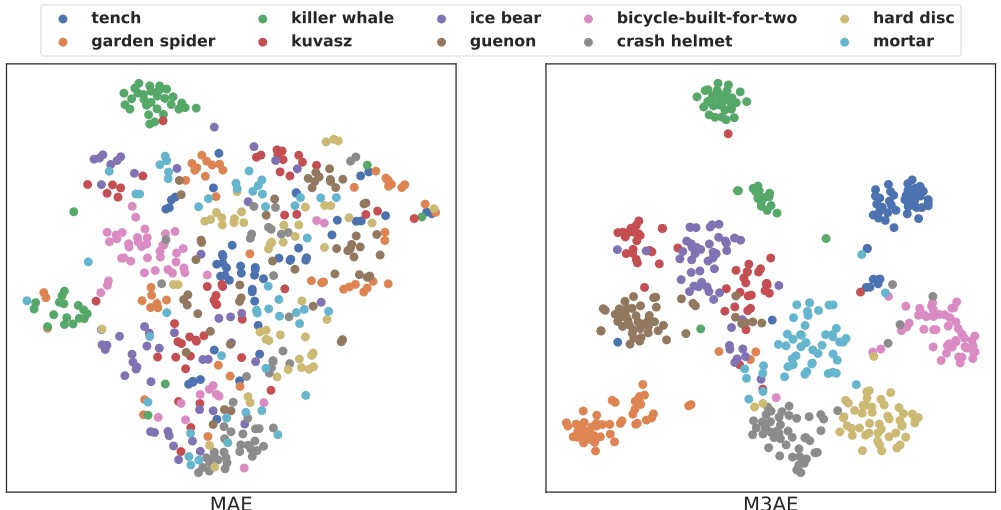

**Figure 11:** t-SNE visualization for learned representations of 10 classes on ImageNet validation set. Left is MAE and right is M3AE. The representation of M3AE clusters much stronger together with the semantic labels compared to MAE representations.

set in Figure 11. Compared to MAE, M3AE successfully clusters together images that correspond to the same semantic label.

## 5 CONCLUSION

In this paper, we propose M3AE, a simple but effective model that learns a multimodal representation from image and language data without the need for contrastive objectives. We show that by pre-training with diverse image and language data, our model can learn shared representations that generalize well to downstream tasks. Due to its flexibility and scalability, M3AE is especially suitable for learning from extremely large-scale datasets, and we envision that such pre-trained models can be broadly applicable in many downstream tasks, such as visual reasoning (Ding et al., 2021), dialog systems (Alayrac et al., 2022) and language guided image generation (Ramesh et al., 2021; 2022).

One major limitation of this work is that the large-scale pre-training of M3AE consumes significant amount of energy. We hope that with the rapid progress of efficient hardware and renewable energy sources, this limitation can be overcome in the near future. Like any other general purpose representation learning models, our model could have both positive (*e.g.*, enabling access to more information by improving machine translation) and negative (*e.g.*, loss of jobs with more automation) impact on the society. These impacts are broadly applicable to pre-trained models in general and not specific to this work.

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

# A  IMPLEMENTATION DETAILS

## A.1  PRE-TRAINING DATASETS

Conceptual 12M (CC12M)[2] (Changpinyo et al., 2021) contains approximately 12M of image-text pairs, the original dataset images are provided in the form of internet URLs. Note that due to some expired URLs and non-English captions, we did not obtain the complete data in the dataset.

## A.2  DOWNSTREAM DATASETS

We evaluate the image encoder transferability on ImageNet (Russakovsky et al., 2014). We report top-1 validation accuracy of a single 256×256 crop. We evaluate evaluate out-of-distribution detection on CIFAR-100 and CIFAR-10 datasets (Krizhevsky et al., 2009). Table 1 provides the detailed information of these datasets.

| DATASET | Classes | Train size | Test size | Evaluation metric |
|---|---|---|---|---|
| CIFAR10 | 10 | 50,000 | 10,000 | Accuracy |
| CIFAR100 | 100 | 50,000 | 10,000 | Accuracy |
| ImageNet | 1000 | $1,281,167$ | 50,000 | Accuracy |

**Table 1:** Details of downstream datasets

## A.3  NETWORK ARCHITECTURES

Following MAE, we use ViT (Dosovitskiy et al., 2020) as the model architecture and consider three different sizes of ViT for the M3AE image and text encoder. The model consists of a stack of standard Transformer blocks (Vaswani et al., 2017), and each Transformer block consists of a multi-head self-attention and an MLP. We use the original ViT-B/16 and ViT-L/16 architectures (Dosovitskiy et al., 2020) for our encoder, as well as ViT-S/16 (Touvron et al., 2021) which is comparable to ResNet-50 in FLOPs and parameters. Following MAE (He et al., 2021), our decoder is lightweight and has 8 blocks and a width of 512. As in MAE, since our encoder and decoder have different width, we adopt a linear projection layer after the encoder to match the dimension. For linear probing, we use the auxiliary CLS token for training the classifier as done in MAE.

## A.4  PRE-TRAINING HYPERPARAMETERS

For the pre-training of M3AE and MAE, we follow the hyperparameters of the original MAE. We keep the optimizer, learning rate, weight decay the same as the original MAE on ImageNet. The only additional hyperparameters unique to M3AE are text token mask ratio and text token classification loss weight. We provide all the hyperparameters in Table 2, where the same hyperparameters are used to train network of all sizes and epochs. The base learning rate corresponds to the learning rate of 256 batch size, and it is linearly proportionally scaled according to the actual batch size.

## A.5  DOWNSTREAM EVALUATION HYPERPARAMETERS

For downstream tasks of linear classification on ImageNet and OOD detection on CIFAR, we use the same hyperparameters for M3AE and MAE. We list the hyperparameters for ImageNet 1K linear classification in Table 3, and OOD detection for CIFAR in Table 5

## A.6  COMPUTATION RESOURCES

All the experiments are performed on the Google Cloud TPU platform. We implement our model using JAX and parallelize the large batch training across many TPUs with data parallelism. For all the pre-training, we use batch size 4096. We report the total amount of compute and the type of resources used in Table 6.

---

[2]https://github.com/google-research-datasets/conceptual-12m

| Hyperparameter | M3AE | MAE |
|---|:---:|:---:|
| Optimizer | AdamW | |
| Base learning rate | 1.5e-4 | |
| Weight decay | 0.05 | |
| Optimizer momentum | $\beta_1 = 0.9, \beta_2 = 0.95$ | |
| Batch size | 4096 | |
| Learning rate schedule | cosine decay | |
| Warmup epochs | 5 | |
| Image data augmentation | RandomResizedCrop | |
| Image patch mask ratio | 0.75 | |
| Text token mask ratio | 0.75 | N/A |
| Text token cross entropy loss weight | 0.5 | N/A |

**Table 2:** Hyperparameters for pre-training M3AE and MAE on CC12M

| Hyperparameter | M3AE and MAE |
|---|---|
| Optimizer | LARS |
| Base learning rate | 0.1 |
| Weight decay | 0 |
| Optimizer momentum | 0.9 |
| Batch size | 2048 |
| Learning rate schedule | cosine decay |
| Epochs | 90 |
| Warmup epochs | 10 |
| Image data augmentation | RandomResizedCrop |

**Table 3:** Hyperparameters for linear classification on ImageNet 1K

| Hyperparameter | M3AE and MAE |
|---|---|
| Optimizer | AdamW |
| Base learning rate | 0.001 |
| Weight decay | 0.05 |
| Optimizer momentum | $\beta_1 = 0.9, \beta_2 = 0.999$ |
| Batch size | 1024 |
| Learning rate schedule | cosine decay |
| Epochs | 50 |
| Warmup epochs | 5 |
| Image data augmentation | RandAugment |
| Label smoothing | 0.1 |
| Drop path | 0.1 |

**Table 4:** Hyperparameters for fine tuning on ImageNet.

| Hyperparameter | M3AE and MAE |
|---|---|
| Optimizer | AdamW |
| Base learning rate | 0.001 |
| Weight decay | 0.05 |
| Optimizer momentum | $\beta_1 = 0.9, \beta_2 = 0.999$ |
| Batch size | 1024 |
| Learning rate schedule | cosine decay |
| Epochs | 100 |
| Warmup epochs | 10 |
| Image data augmentation | RandAugment |

**Table 5:** Hyperparameters for fine tuning on CIFAR10.

| Model | ViT-S | ViT-B | ViT-L |
|-------|-------|-------|-------|
| MAE | 16.5h (v3-64) | 8.5h (v3-128) | 11.5h (v3-128) |
| M3AE | 9.5h (v3-128) | 5h (v3-256) | 10h (v3-256) |

**Table 6:** TPU pod size and compute hours used for training 50 epochs of M3AE and MAE on CC12M.

