# OpenReview forum: "Multimodal Masked Autoencoders Learn Transferable Representations"
_ICLR.cc/2023/Conference — Submitted to ICLR 2023_

### Official Review · Reviewer_GbHq · 2022-10-23

**Confidence:** 4
**Correctness:** 2
**Technical Novelty And Significance:** 3
**Empirical Novelty And Significance:** 2
**Recommendation:** 3

**Clarity, Quality, Novelty And Reproducibility:**

Clarity - good; the paper is clear and well-written.

Quality - the experiments needs work. The main experimental result (that M3AE outperforms MAE) is not significant enough (weakness 1), and the other main claims in the introduction are also flawed (weaknesses 2-3)

Originality - ok. The concept of masked auto-encoding is not novel, although extension to the multimodal setting merits some consideration.

**Strength And Weaknesses:**

Strengths: The paper is easy to understand, with the methodology clearly presented. The results section includes some nice qualitative analysis and visualizations to help understand the representations that are learned by M3AE.

Weaknesses:

1) The empirical support for the proposed approach is weak.

One of the main takeaways from the experiments (Figure 3-5) is that M3AE outperforms MAE in the downstream ImageNet classification task. This is a good sanity check, but it is not significant nor surprising. The M3AE framework involves predicting text token logits from image patches, which is similar to a supervised classification task where the text is providing labels for training the image encoder. On the other hand, MAE is completely unsupervised.

It would be better to compare M3AE against other vision-language pretraining methods. However, no vision-language pretraining baselines are provided except for CLIP which outperforms M3AE by a large margin in Figure 3.

2) In the introduction, the authors make the strong claim that M3AE offers two benefits over contrastive pretraining methods, but neither benefit is demonstrated in the experiments.

First, the authors emphasize that contrastive learning "requires paired image-and-text data and therefore cannot leverage widely available unpaired data." This suggests that leveraging unpaired data will make M3AE perform better than CLIP. But as shown in Figure 3, CLIP does much better than M3AE. To support this claim, the authors should either (i) decrease the paired data for CLIP to e.g., 10-30% and show that M3AE does better than CLIP for the same amount of paired data in Figure 3, or (ii) add some unpaired image or text data for M3AE and show that it can do better than CLIP by leveraging this unpaired data.

Second, the authors claim that the "separation of image and text encoders" in contrastive learning "hinders the joint understanding of image and text", but they provide no proof of this. To support this claim, they should show that M3AE performs better than contrastive learning on a multimodal downstream task. Also, note that separation of image and text encoders is not necessary in contrastive learning (e.g., GLIPv2 enables fusion between image and text features before the contrastive objective), so this is not an inherent limitation of contrastive pretraining methods.

3) The authors write: "Surprisingly, we find that M3AE benefits from a higher text mask ratio (50-90%), in contrast to BERT whose standard masking ratio is 15%." This statement is misleading because it suggests that M3AE benefits from a higher text mask ratio compared to BERT on the same downstream task. But actually, BERT embeddings are evaluated on language tasks while M3AE embeddings are evaluated on image tasks. It does not make sense to compare BERT's text masking ratio with M3AE's, since M3AE's text representation is never evaluated in a downstream task.


**Summary Of The Paper:**

This paper proposes an extension of Masked Autoencoders (MAE) to the vision-language domain called Multi-Modal Masked Autoencoder (M3AE). Image-caption pairs are partially corrupted by masking some image patches and some language tokens. The non-masked image patches and language tokens are passed through an autoencoder based on transformers, which is trained to reconstruct the full image-caption, including masked parts. The embeddings of image patches that are learned by the transformer encoder are transferred to downstream image tasks, such as classification and out-of-distribution detection.

**Summary Of The Review:**

See weaknesses 1-3 above.

---

> ### Author Response · Authors · 2022-11-19
> **Author Response**
>
> First of all we’d like to thank the reviewer for the detailed and constructive comments. We address the specific questions below.
>
>
> > It would be better to compare M3AE against other vision-language pretraining methods. But as shown in Figure 3, CLIP does much better than M3AE. To support this claim, the authors should either (i) decrease the paired data for CLIP to e.g., 10-30% and show that M3AE does better than CLIP for the same amount of paired data in Figure 3, or (ii) add some unpaired image or text data for M3AE and show that it can do better than CLIP by leveraging this unpaired data.
>
> We’d like to thank the reviewer for the great suggestions on these experiment setups. We want to clarify that the CLIP results in this paper are directly taken from a baseline in a prior work [1], and we have not trained CLIP on this dataset ourselves. We agree with the reviewer that it would be important to perform a more apple-to-apple comparison with CLIP in the standard as well as the unpaired setting. Due to the limitation in computation resources, we are unable to complete such experiments during the response period. We will include the results in the next version of the paper as soon as they are ready.
>
> > To support this claim, they should show that M3AE performs better than contrastive learning on a multimodal downstream task. Also, note that separation of image and text encoders is not necessary in contrastive learning (e.g., GLIPv2 enables fusion between image and text features before the contrastive objective), so this is not an inherent limitation of contrastive pretraining methods.
>
> We’d like to thank the reviewer for suggesting these tasks to evaluate the multi-modal capability of our method. However, given the limited compute resources we have access to, we are unable to complete them during the response period. We will include them in the next version of the paper.
>
>
>
> > This statement is misleading because it suggests that M3AE benefits from a higher text mask ratio compared to BERT on the same downstream task.
>
> We fully agree with the reviewer that the task here is different from the NLP tasks that BERT is evaluated from, so this is not a truly apple-to-apple comparison. We believe that using such a high mask ratio is still surprising to us despite the task difference because language encoders in prior works don’t usually use such a high mask ratio, so we expected similar trends should still hold here.
>
>
> We hope our response can address the reviewer's concerns.

---

### Official Review · Reviewer_1MQ5 · 2022-10-24

**Confidence:** 3
**Clarity, Quality, Novelty And Reproducibility:** 1. Clarity - good; the paper is well-…
**Correctness:** 3
**Technical Novelty And Significance:** 2
**Empirical Novelty And Significance:** 2
**Recommendation:** 3

**Strength And Weaknesses:**

Strength:

1. The paper is well written and this reviewer enjoyed reading it.

2. The proposed approach is simple and intuitive.

3. It shows promising results on image classification and out-of-distribution detection tasks.

Weakness:
1. The idea lacks of novelty.

2. No experiments for vision-language tasks have been performed (e.g., text-to-image retrieval, VQA, visual reasoning, etc). The authors only fine-tune their pre-trained models for visual tasks, such as out-of-distribution detection and image classification. As the paper targets at multi-modal pre-training, it's essential to provide the experiments for vision-language down-stream tasks!

3. Figure 2 is confusing. For example, in the lower left corner, the text "Trees in a winter storm" even appears in the space of its right image.

4. Ablation study experiments aren't sufficient, it's important to conduct the experiments, e.g., different mask ratios of MAE.

5. Why not use two separate encoders for visual and textual feature extraction? The proposed M3AE only use one encoder for these two kinds of modalities together, but this may make the models confused. it's better to perform ablation study to use two different encoders, so that we can figure out the necessity of such design.

6. The comparison is unfair, the proposed method uses more vision-language data (e.g. CC12M) for pre-training, and compare the models with MAE, which is only pre-trained on ImageNet. The performance gap between M3AE and CLIP is still very large.

**Summary Of The Paper:**

This paper presents an empirical study of M3AE trained on a large-scale image-text dataset, and find that multimodal masked autoencoder is able to learn generalizable representations that transfer well to downstream tasks.

**Summary Of The Review:**

The paper is about the new approach for pre-training vision-language models by the proposed multimodal masked autoencoder. Although the approach is interesting, it is incremental and not novel. Some of the experiments demonstrating the superiority of the proposed approach are also unclear.

The proposed M3AE haven't performed experiments on vision-language down-stream tasks, it's not clear that the approach can learn good multimodal representation. Although the model can achieve better performances compared with MAE, M3AE uses larger dataset (e.g. CC12M). The performance gap between M3AE and CLIP is large.

Some important ablation study experiments, such as different mask ratios of MAE and different encoders for visual/textual inputs, aren't conducted.

---

> ### Author Response · Authors · 2022-11-19
> **Author Response**
>
> First of all we’d like to thank the reviewer for their detailed and constructive comments. We address the specific questions below.
>
>
> > No experiments for vision-language tasks have been performed.
>
> We’d like to thank the reviewer for suggesting these tasks to evaluate the multi-modal capability of our method. However, given the limited compute resources we have access to, we are unable to complete them during the response period. We will include them in the next version of the paper.
>
> > Ablation study experiments aren't sufficient, it's important to conduct the experiments, e.g., different mask ratios of MAE.
>
> The mask ratio of MAE has been extensively studied in the original paper, and 75% is the reported optimal mask ratio. Hence, in this paper, we directly follow the results of MAE and use the 75% mask ration. For the additional text mask ratio we introduced in the paper, we have conducted ablation studies and determined the optimal text mask ratio is also 75%.
>
>
> > Why not use two separate encoders for visual and textual feature extraction?
>
> Our model follows the paradigms of masked autoencoders and is trained using the masked token prediction loss instead of the contrastive loss. Using two separate encoders for images and text prevents the two encoders preventing the self-attention layers of each modality from attending to other modalities, therefore preventing the information to be shared between the text encoder and image encoder. Since our model is trained using the masked token prediction loss, it is crucial for the encoder to learn to leverage the unmasked image information to predict the text, and vice versa. Finally, using a single encoder significantly simplifies the method, making it much more scalable.
>
>
> > The comparison is unfair, the proposed method uses more vision-language data (e.g. CC12M) for pre-training, and compares the models with MAE, which is only pre-trained on ImageNet.
>
> We note that the MAE baselines used in this paper are pre-trained on the same dataset of CC12M instead of ImageNet. All the MAE models are trained for the same number of epochs on CC12M using the same hyperparameters as M3AE for a fair comparison.
>
>
> We hope our response can address the reviewer's concerns.

---

### Official Review · Reviewer_qu2E · 2022-10-25

**Confidence:** 4
**Correctness:** 3
**Technical Novelty And Significance:** 2
**Empirical Novelty And Significance:** 2
**Recommendation:** 5

**Clarity, Quality, Novelty And Reproducibility:**

The presentation is clear to me. and I'm satisfied with the details of the technical points. I think this work can be easily reproduced. Regarding novelty, I'm ok with the proposed method to extend MAE to multimodal representation learning. While I like to see the clear comparison with other existing efforts like BEIT-3 with a similar motivation.

**Strength And Weaknesses:**

Strength

-The idea of applying masked autoencoder strategy to multimodality data is rational. The finding on higher mask ratio on text performs better than the low mask ratio on pure text is interesting. The explanation makes sense to me.

-The experiments show the strength over MAE in different scenarios and applications.



Weakness

-There is other work also working on the similar motivation, i.e., leveraging unpaired data and paired text-image data for multimodal representation learning. BEIT-V3 [Wang et al.,], where they demonstrate an exceptional performance over many existing solutions including CLIP on a number of downstream tasks. It seems this paper misses the discussion on this work. And I’d also like to see the experimental comparison with BEIT-3 to show the advantages.

[Wang et al.,] Wang et al.,Image as a Foreign Language: BEiT Pretraining for All Vision and Vision-Language Tasks, https://arxiv.org/abs/2208.10442

-I have a little concern on the effectiveness on the large-scale data, such as several hundreds of millions of data points or even larger. The current training is on CC-12M, and shows a good improvement over MAE. However, when the number of training samples gets larger, I am not sure if this improvement margin can be preserved.

-Following the above point, thanks to the effort on the large scale data curation LAION, there are billions of image –text noisy pairs available. Please explain the application scenario of this method.

-Figure 9, the reconstruction result from MAE is missing. I’d like to see a comparison to show the strength.

-Other minors: In page 3, “For patches and tokens, we sample s random”, not sure what “s” means here.

In page 7, there is a space in “M3AEoutperforms”

**Summary Of The Paper:**

This paper studies the multimodal representation learning in the masked auto-encoder way. The work is motivated by the fact that many existing multimodal learning approaches require a large amount of paired text-image data. This work builds upon MAE, extending it to text modality as well. The proposed method can handle both the unpaired image data and paired image-text data. The experiments show a clear improvement over MAE in several tasks, and show the effectiveness of the learned transferrable representation.

--- post-rebuttal ---\
Thanks for the authors' response. I have carefully read the response and other reviewers' comments. I believe the current form of this paper  is not ready for ICLR publication. I keep my score.

**Summary Of The Review:**

Overall, the proposed solution is rational and mostly well presented. While my major concerns are lack of comparison with the existing method and no clear motivation given billions of paired text image data available.

---

> ### Author Response · Authors · 2022-11-19
> **Author Response**
>
> First of all we’d like to thank the reviewer for their detailed and constructive comments. We address the specific questions below.
>
> > Comparison to BEIT-v3
>
> Thanks for pointing out this related work. We note that due to the relative short gap between the ICLR deadline and the release of BEIT-v3 (a month), we believe that our work should be considered as concurrent to BEIT-v3. Also since the BEIT-v3 model from the original paper is trained on a larger dataset than M3AE and the BEIT-v3 authors have not released the code and pre-trained model, it is difficult to compare at this moment. We are planning to include the comparison to BEIT-v3 once the implementation or pre-trained models are released.
>
> > Effectiveness on the large-scale data such as LAION
>
> We agree with the reviewer that it would be important to pre-train our model on a larger scale dataset such as LAION. However, training such a model is very computationally expensive, and given the amount of compute resources we have access to now, we are unable to train such a model during the response period. We are planning to include the results in the next version of the paper.
>
> > Figure 9, the reconstruction result from MAE is missing.
>
> Figure 9 aims at qualitatively evaluating the reconstruction of our model to ensure that M3AE makes semantically coherent reconstructions. We will include the MAE reconstructions in the next version of the paper.
>
>
> We hope our response can address the reviewer's concerns.

---

### Official Review · Reviewer_ZSrM · 2022-10-27

**Confidence:** 3
**Clarity, Quality, Novelty And Reproducibility:** Well-motivated paper yet lacks discus…
**Correctness:** 2
**Technical Novelty And Significance:** 3
**Empirical Novelty And Significance:** 2
**Recommendation:** 5

**Strength And Weaknesses:**

Strengths:
(a) The paper is overall well-written, the problem well formulated and novel to me. Generalizing masked autoencoders to work on more modalities is definitely interesting to the community.
Weaknesses:
(a) There seems to be a lack of comparison with other baselines. The paper only compare with CLIP and MAE, while not other baselines, such as MultiMAE[1], using the same modality-agnostic setting, were not included.
(b) The author should also include a discussion on how is M3AE different from MultiMAE.
[1] MultiMAE: Multi-modal Multi-task Masked Autoencoders, ECCV 2022

**Summary Of The Paper:**

The paper scales masked autoencoders (MAE) to image-text multimodal data, it investigates the use of modality-agnostic unified encoders, which learns generalizable representations across a number of downstream tasks.

**Summary Of The Review:**

The paper is overall well-motivated and well-written, yet the experiment part seems to fall short with a lack of discussion and comparison with other baselines.

---

> ### Author Response · Authors · 2022-11-19
> **Author Response**
>
> First of all we’d like to thank the reviewer for their detailed and constructive comments. We address the specific questions below.
>
> > Comparison to MultiMAE
>
> Thanks for pointing out an interesting connection between our method and MultiMAE. Despite the similarity in names, our method and MultiMAE apply to different settings and therefore cannot be directly compared. MultiMAE learns a joint representation for RGB image, depth map and semantic segmentation, while our method learns the representation for natural image and language. Because of this major difference between the data modalities M3AE and MultiMAE are learning from, it is difficult to directly compare these two methods. Furthermore, we note that for the CC12M image-text dataset we use in this paper, depth and semantic segmentation data is not available, and therefore MultiMAE cannot be trained on this dataset.
>
>
> We hope our response can address the reviewer's concerns, and we kindly ask the reviewer to participate in the discussion.

---

### Decision · Program_Chairs · 2023-01-20

**Decision:**

Reject

**Justification For Why Not Higher Score:**

1. Similar ideas have been investigated previously, namely leveraging unpaired data and paired text-image data for multimodal representation learning.
2. Additional experiments and comparisons on vision-language tasks should be conducted to verify the effectiveness of the proposed approach.
3. The paper has some degree of scientific novelty, but may not meet ICLR's very high standards.

**Justification For Why Not Lower Score:**

1. The paper presents a novel Multimodal Masked Autoencoder (M3AE) model that learns a unified encoder for both vision and language data via masked token prediction.
2. The generalization of masked autoencoders to work on multiple modalities is novel and has proven effective.

**Metareview: Summary, Strengths And Weaknesses:**

1. The paper presents a novel Multimodal Masked Autoencoder (M3AE) model that learns a unified encoder for both vision and language data via masked token prediction.
2. The generalization of masked autoencoders to work on multiple modalities is novel and has proven effective.
3. Similar ideas have been investigated previously, namely leveraging unpaired data and paired text-image data for multimodal representation learning.
4. Additional experiments and comparisons on vision-language tasks should be conducted to verify the effectiveness of the proposed approach.
5. The paper has some degree of scientific novelty, but may not meet ICLR's very high standards.

**Summary Of Ac-Reviewer Meeting:**

The scores from the reviewers are quite consistent. Although the authors have replied to several comments, the reviewers still think the paper could not meet the very high standard of ICLR. Therefore, there was no AC-reviewer meeting for this paper.